# Biomimetics Linked to Classical Product Development: An Interdisciplinary Endeavor to Develop a Technical Standard

**DOI:** 10.3390/biomimetics7020036

**Published:** 2022-03-30

**Authors:** Kristina Wanieck, Leandra Hamann, Marcel Bartz, Eike Uttich, Markus Hollermann, Manfred Drack, Heike Beismann

**Affiliations:** 1Faculty of Applied Informatics, Deggendorf Institute of Technology (DIT), Teaching Area Biomimetics and Innovation, Grafenauer Str. 22, 94078 Freyung, Germany; kristina.wanieck@th-deg.de; 2Institute of Evolutionary Biology and Animal Ecology, University of Bonn, An der Immenburg 1, 53121 Bonn, Germany; lhamann@evolution.uni-bonn.com; 3Engineering Design, Faculty of Engineering, Department of Mechanical Engineering, Friedrich-Alexander-University Erlangen-Nürnberg (FAU), Martensstraße 9, 91058 Erlangen, Germany; bartz@mfk.fau.de; 4Product Development, Institute Product and Service Engineering, Faculty Mechanical Engineering, Ruhr-University Bochum (RUB), Universitaetsstr. 150, 44801 Bochum, Germany; uttich@lpe.rub.de; 5Die Bioniker GbR/ELISE GmbH, Consulting & Development, Im Mersch 14, 49577 Eggermühlen, Germany; markushollermann@diebioniker.de; 6Evolutionary Biology of Invertebrates, Institute of Evolution and Ecology, University of Tübingen, Auf der Morgenstelle 28E, 72076 Tübingen, Germany; manfred.drack@uni-tuebingen.de; 7Department of Mechanical Engineering, Westphalian University of Applied Sciences, Teaching Area Biology and Biomimetics, Münsterstr. 265, 46397 Bocholt, Germany

**Keywords:** innovation, product development, process, solution finding, workflow

## Abstract

Biomimetics is a well-known approach for technical innovation. However, most of its influence remains in the academic field. One option for increasing its application in the practice of technical design is to enhance the use of the biomimetic process with a step-by-step standard, building a bridge to common engineering procedures. This article presents the endeavor of an interdisciplinary expert panel from the fields of biology, engineering science, and industry to develop a standard that links biomimetics to the classical processes of product development and engineering design. This new standard, VDI 6220 Part 2, proposes a process description that is compatible and connectable to classical approaches in engineering design. The standard encompasses both the solution-based and the problem-driven process of biomimetics. It is intended to be used in any product development process for more biomimetic applications in the future.

## 1. Introduction

Using nature as a model, biomimetics has the potential to optimize products, save resources, or even lead to disruptive innovations that can induce a paradigm shift [1]. The number of scientific publications in the field of biomimetics has significantly increased from less than one hundred in the early 1990s to more than 2500 in 2017 [2,3]. In an unspecific search using only the search term “biomimetics,” more than 10,000 documents can be found in scientific databases and more than 100,000 in Google Scholar [4]. Success stories are found from diverse fields, ranging from communication and sensory systems using evolutionary strategies and optimization of materials, to architecture and design, as well as fluid- and aerodynamics [1]. Additionally, due to the large number of species and their morphological diversity, it is considered that “researchers pursuing biomimetic approaches have barely scratched the surface of biological inspiration” [5], and the estimated potential of biomimetics for innovation is high. 

However, the practice of biomimetics, i.e., product development based on a biological model progressing up to market-ready products, is rarely carried out. According to the BioM Innovation Database, when published in 2014, 86 commercially available products met the criteria for being considered as biomimetic, meaning that they mimic a biological function, its forms, processes, or interactions [6]. The criteria of the International Standardization Organization (ISO) in ISO 18458 might even be stricter [7]. Thus, the authors assume that biomimetics, in the strict sense of ISO 18458 [7] has not yet established itself as a recognized method for product development and engineering design in practice, especially in industrial procedures, due to the lack of (1) awareness, (2) an available process description, and (3) a measurable and robust benefit for the company. 

Based on the experience of the authors, biomimetics can offer solutions for industry in three different ways: (1) Design principles and concrete ideas that are close to the market, i.e., the biological function is clear, and its underlying working principle is easy to implement; (2) Transformational ideas, of which the biological function is clear, but its transfer and application is unclear, and (3) Research actions, in which a biological model is not yet identified, or a technical application has yet to be found. In all three cases, a general biomimetic process description is required that is adaptable and applicable to specific research questions and development projects. Moreover, the biomimetic process still exhibits various challenges in theory, research, and practice [8,9,10]. One of these challenges is the clear understanding of what needs to be done and how this can be achieved during the biomimetic process. In order to address this challenge, several research endeavors have focused on process descriptions, e.g., the work of [11,12] and its improvements, the usage of tools and methods to support industry adoption [13], or to consider the perspective of practitioners [14]. 

Due to the large variety of applications, descriptions of biomimetic approaches are often idealized and general, as seen in the international standard on biomimetics ISO 18458—terminology, concepts, and methodology [7]—or in the unified problem-driven approach [12]. ISO 18458 even states that it was impossible to find one process description that fits every project [7]. This aspect is in contrast to industrial practice, where various methodologies are used to develop different technical products, such as the domain-independent approaches, according to Pahl and Beitz [15]. In Germany, it is common to use technical standards in engineering processes, such as those developed by the Association of German Engineers (Verein Deutscher Ingenieure, VDI), like the standard VDI 2221, for example, on the design of technical products and systems [16,17], or domain-specific standards, such as VDI 2206, for the development of mechatronic products [18]. 

Concerning biomimetics, technical standards have been developed by the VDI for terms and definitions, as well as on specific topics such as lightweight design and optimization. These standards have become the foundation for the three international standards on biomimetics [7,19,20], which indicates the importance of standardization work on a national level. Interestingly, the standards of biomimetic methods have proven to facilitate product developments. Both international standards and technical standards are already used during biomimetic developments, as they are mentioned in three patents and in 49 scientific publications [4]. However, only a few approaches describe how biomimetic methods fit in specific technical processes, e.g., in the context of structural optimization processes in lightweight design [21] or for motion control in robotics [22]. A general linkage of biomimetic methods and general engineering methodologies has only taken place as first attempts [12,23]. Therefore, to increase the innovation potential of biomimetics, biomimetic methods need to be considered in engineering design, independent of specific existing development processes used in the respective companies. 

The authors assume that the missing link between biomimetic and established, classical product development methodologies prevent engineers from diving deeper into biomimetics and thus could also prevent the successful implementation of biomimetics in the product development process. Additionally, a standard can help to increase the plannability and estimate potential risks. Once these aspects become more predictable, the acceptance of biomimetics might increase [23,24]. Therefore, several strategies have been identified to overcome these barriers and increase the number of market-ready products, i.e., (1) to describe an approach that can be adapted to individual projects [12], (2) to support users with methods and tools [23,25,26], and (3) to link this approach to established working procedures already used in industry. 

In 2019, the VDI invited experts from various scientific disciplines and industry to collaborate and focus on step 3, i.e., to develop a technical standard that explains how to integrate biomimetics in classical product development processes, with step-by-step guidance for industrial practice. Biomimetics is defined as the “interdisciplinary cooperation of biology and technology or other fields of innovation with the goal of solving practical problems […]” [7]. As such, it can be understood and performed in various ways, e.g., focusing on technical functional improvements and/or considering efficiency strategies for biological systems, leading to more sustainable products. Furthermore, regarding the biomimetic process, there are similarities or links to other design approaches or management activities, such as TRIZ, Design Thinking, or Circular Economy, which could have also been linked to biomimetics. 

In this article, a smaller group from this expert panel describes the workflow towards the new VDI Standard, VDI 6220 Part 2—Biomimetic Development Process; Products and Procedures -, which is available in German, but will soon be translated to English. The aim of this article is to present the efforts to standardize a pragmatic approach within the context of VDI standards. As such, it will help the scientific community to better understand the outcome of this work, i.e., the technical standard, and for anyone interested in the background information. It is the only article in the field that describes the standardization process and that emphasizes the link to classic product development processes. Therefore, the manuscript opens the scientific discussion in the field about the characteristics of biomimetics and other approaches, leading to a foundation for future work and the transformation of biomimetics.

The authors want to contribute to the development of a repeatable and robust methodology, which is considered to be a scientific challenge [27]. Since the standardization work is not known in its entirety in the academic field, the general principles of standardization work in the VDI, based on the VDI 1000—VDI Standardization Work—Principles and Procedures, are also explained here. The authors consider this background information to be of value to the scientific community, as similar endeavors might take place in other countries. Finally, the article presents the integration of biomimetic methods into existing product development methodologies and facilitates access to biomimetic methods by practitioners such as designers, engineers, industrial engineers, and product managers.

## 2. Materials and Methods

### 2.1. Expert Panel for VDI 6220 Part 2

According to VDI 1000, the expert panel for VDI 6220 Part 2 included all relevant interested parties. Box 1 shows the main principles for developing a technical standard and setting up the panel. The standard committee consisted of 21 members from science, research, and industry, a secretary of the VDI, and one guest from a networking association, all of whom combined their expertise to establish an approach for the development of biomimetic products. Figure 1 shows the participants of the expert panel. According to their employment, five members were assigned to industry and 16 to science and research. Based on their field of expertise, eight belong to the engineering sciences, six have a background in biology, and seven have expertise in interdisciplinary fields such as biomimetics itself, or work between engineering sciences and natural sciences. Some of the members could be allocated to more than one field of expertise, but only their primary interest was counted.

Box 1Procedure for developing a technical standard based on VDI 1000.
**VDI 1000—VDI Standardisation Work—Principles and Procedure**
Anyone can submit proposals for a new VDI guideline.Based on specified criteria it is checked whether an initiation of standardisation work is justified, i.e.,(a) whether there is a current demand, (b) whether the topic is already covered, and (c) whether enough people from interested
parties are willing to participate on a voluntary basis and contribute with their expertise. VDI standardization work serves society as a whole and is not intended to provide individuals an economic advantage. Individual members represent their personal expert perspective or reflect the perspective of an interested party.Appointing committee members is done carefully to ensure that all relevant interested parties are represented adequately. Committees develop preliminary, internal working papers for the purpose of preparing the manuscript oft he guideline. Decisions should be made by consensus; if this is not possible, a vote must be taken. Consensus means: “general agreement, characterized by the absence of sustained opposition to substantial issues by any important part of the concerned interests and by a process that involves seeking to take into account the views of all parties concerned and to reconcile any conflicting arguments [DIN EN 45020, 1.7]”. Whereby “Consensus need not imply unanimity. [DIN EN 45020, 1.7]”. With the publication of the draft, the general public can exert influence during the public inquiry procedure. Objections of the public are processed in the committee and, decisions are made by consensus or resolution and are documented. The guideline is then published and replaces the draft.

The committee was founded on 16 May 2019 in Düsseldorf, Germany. Overall, eleven meetings have led to a consensus concerning a general biomimetic development process description, and the group adopted the VDI standard draft on 21 September 2021. This VDI standard draft describes the methodological procedure recommended for a biomimetic development process. With the publication of the draft, the public can comment and advise on the public inquiry procedure. A draft is not considered a generally recognized rule for engineering workflows, but it can show trends of future technical progress. The public inquiry procedure starts with the publication of the draft (expected June 2022) and runs for four months. The received objections are reviewed by the committee and, in accordance with the principles of VDI standardization work, decisions are made by consensus or resolution and are then documented. The standard is then published, replacing the draft.

### 2.2. Standardization Work

In addition to the panel’s expertise and their knowledge of the state of the art, the new standard could draw on all previously published VDI standards on biomimetics (see http://www.vdi.de/bionik; accessed on 23 February 2022) and incorporate the experience gained from these drafting processes. Furthermore, some panel members are also active on other standardization committees dealing with biomimetics, such as DIN (Deutsches Institut für Normung, the German Institute for Standardization) and ISO (the International Standardization Organization). Thus, experience from the international committees and participating communities could also be incorporated into the development of VDI 6220 Part 2. Figure 2 summarizes the standardization working process.

From the beginning of the work on VDI 6220 Part 2, special emphasis was placed on the fact that biomimetics should be understood as an addition to existing development processes; thus, it addresses engineers and practitioners who use technical standards. In German-speaking countries, VDI 2221 Part 1 is a proven standard used in many development processes. Hence, it was selected to develop a linkage to VDI 6220 Part 2. However, complementary conceptualizations in the context of biomimetics and product development have also been discussed and considered [18,23,25,26,28,29].

The production process, i.e., the discursive preparation of a standard draft, is comparable to an open review process in which all members read and correct the texts, if necessary. As soon as the working papers have reached a certain level of maturity, they are further edited so that the finished manuscript corresponds to the consensus found by the committee, thus complying with the state of the art after the public inquiry procedure. The content of a standard can therefore no longer be traced back to specific authors, and all content and working documents are to be treated confidentially until the draft stage. In addition, a standard must be formulated as a very compact and practicable recommendation for action. The production process, in particular the discussions that led to the result of a technical standard and thus to the state of the art, eludes scientific scrutiny, and only the condensed result is reported.

### 2.3. Common Language for the Biomimetic Approaches

A general recommendation in standardization work is to agree on a common language accepted by all panel members. Therefore, a significant amount of committee work was devoted to selecting and defining technical terms that are used accurately in the standard. In this case, knowledge and experience from the use of the already established VDI 2221 Part 1 and Part 2, VDI 2206, and VDI 6220 Part 1 have been incorporated.

As part of the initial definitions, the introduction to VDI 6220 Part 2 defines specific biomimetic terms based on the expert panel’s knowledge and the state of the art. To achieve a common language understandable between biomimetics and classical product development, terms and definitions from, e.g., VDI 2221 and VDI 6220-1 were used for the description of the biomimetic development process. As such, readers will be familiar with the wording and the explanation of the different steps of the biomimetic development process is clear and easy to follow.

## 3. Results

The main achievement of the interdisciplinary work of the VDI 6220 Part 2 committee is the linking of biomimetics with the classical product development process. Figure 3 shows the general structure of this interconnection. Further details can be found in VDI 6220 Part 2 (draft 2022), as copyright issues do not allow reprinting content of a standard. The standard introduces biomimetics as an iterative, 9-step process and offers utilization methods in different phases of the research and development process.

Different options were discussed in the expert panel to illustrate the biomimetic approaches of problem-driven and solution-based methods, ranging from simplifying sequential representations, to using a clock model or a spiral, [24], e.g., in VDI 2221 Part 1 and VDI 2221 Part 2. While sequential representations are often easier to understand, they lack a distinct illustration of the iterative and parallel work in practice that is common for biomimetic working processes. Complex representations illustrate iterative and parallel work, but users might not know where to start and what exactly needs to be done. In favor of a low threshold for users with different backgrounds, the expert panel decided to use a spiral to represent the biomimetic approaches, with additional remarks in the text pointing to the possibility of parallel and iterative work during the process. The spiral is presented to emphasize iterations similar to the V-Model in VDI 2206, where a subsequent process begins with insights from the previous process [18]. Thus, and in contrast to existing biomimetic process representations, the chosen representation of the process is both detailed and formalized.

The biomimetic spiral is split into four sections with continuous transitions. Each section represents a phase in the biomimetic approaches and contains phase-specific activities. The phases are: arrange (I), analyze (II), abstract (III), and apply (IV) and are similar to other process descriptions of biomimetics [12]. The spiral encompasses both the solution-based and the problem-driven approach of biomimetics. When beginning a problem-driven approach, the starting point is at the top of the product development process, with the link to biomimetic methods at the end of the planning, concept development, and design phase. The starting point for the solution-based approach is in the center of the biomimetic spiral.

### The Classical Product Development Process and Its Link with the Biomimetic Approach

The development process according to VDI 2221 Part 1, with the division into the four product development phases (planning, concept development, implementation in design, and elaboration) as introduced by Pahl and Beitz, was chosen intentionally for the classical product development process [15,16]. This choice is because this methodology is well established in industry, and the development process is generic so that it can be applied to a wide range of products (including processes), regardless of the domain. According to VDI 2221, the general product development process is to go through a problem-solving process from a problem to a possible solution. An essential part of developing technical products is the search for solutions, which is mainly carried out in the concept development phase. In the context of the classical product development process, several methods (for finding solutions in the concept development phase) are already provided to support engineers, e.g., morphological boxes and classification schemes [30,31].

By linking the classical product development process with the biomimetic approaches, the classical development process receives a further branch, represented by the bridge in Figure 3, and connected with the methods and contents from VDI 6220-2. The linkage of the product development process with the problem-driven biomimetic approach begins after the planning phase and ends at the end of the concept development phase (or at the start of the implementation in design phase). This extension of the classical development process to a separate but related biomimetic process allows for the expansion of the solution space through the biological-biomimetic domain, subsequently increasing the successful implementation of these found solutions. Both the classical process and the iterative biomimetic approach benefit from the knowledge of the other, e.g., new insights during the product development process might require another iteration in the biomimetic spiral, while insights during this iteration might trigger additional activities in the product development process.

One prerequisite to begin the overall process of the biomimetic approaches is the willingness to dive into biology, to have a basis of biological know-how [8] or at least to collaborate. Several efforts have already been made to systematize the search for biological solutions. Ontologies and keyword finders help to make the search more precise, as well as databases and programs on the World Wide Web, some of which are freely available, to find suitable models. A listing can be found in VDI 6220 Part 2, Beismann 2018 [23] or Wanieck et al., 2017 [26]. However, all previous approaches have their limitations, either due to the abundance of possible solutions in nature or due to the lack of knowledge about the biological models. The expert panel did not choose specific tools, databases, or methods to include during the process, but they are mentioned in the technical standard. Therefore, it might be necessary to conduct additional biological investigations during the process. This should be encouraged so that the transfer and interdisciplinary technological implementation process is further supported. However, VDI 6220-2 does not provide biological methods to improve the knowledge on the biological system, but primarily proposes methods to transfer existing biological knowledge to engineering problems with interdisciplinary teams—one core step in the biomimetic approach. VDI 6220 Part 2 is composed so that anyone interested in the topic can explore the approaches and its outcomes. The biomimetic approaches clearly benefit from interdisciplinarity and teamwork. Sometimes, bioinspiration comes from unexpected domains that cannot be planned, but which require an open-minded team to be recognized.

In the case of a solution-based approach, the product development process starts from within the spiral. This process is mainly initiated by biologists that collaborate with engineers when entering the product development process.

## 4. Discussion

The new standard VDI 6220 Part 2 links biomimetics to the classical product development process commonly used by engineers. For this purpose, the expert panel agreed that biomimetics, in this case, is understood primarily as a supplement to existing development processes to enable its linkage into already established processes, instead of introducing a new, stand-alone process. This aspect might help reduce inhibitions and increase the usage of biomimetics in technical fields of industry with engineers as the primary users of VDI standards. Hence, it is an addition to the product development process, such as VDI 2221.

Nevertheless, the intensive work that needs to be done during a biomimetic development process remains, along with some of its challenges. Some of these challenges are: (1) Product designs are based on human-made decisions, whereas biological solutions result from evolution. Thus, structural and functional solutions in organisms have design constraints, such as phylogenetic, functional, morphogenetic, and environmental limitations [32]. (2) During the biomimetic working process, the transferability and scalability of the biological model have to be analyzed [33,34]. A body shape that might be advantageous at high Reynolds numbers might not be as efficient at low Reynolds numbers, given that here, viscous forces instead of inertial forces are dominating [32]. (3) Multifunctionality in biological systems might hinder the extraction of a single desired working principle [24], or biological principles might lose their functionality when applied to different scale dimensions [32]. (4) On the technical side, feasibility might be limited through the current state of manufacturing processes, such as the production of hierarchically organized materials or dynamic, sensor-integrated systems. Finally, (5) biomimetic developments might take longer than classical ones, with an average development time of 6–9 years [1].

Even though biomimetics has been described as an additional method for an engineering process within the new VDI standard, it can also be described as an individual field of research. It is still possible to follow the biomimetic process on its own as an essential method for solution-finding, especially with the solution-based approach. Even if the difference of how to use biomimetics may seem to be slight, using biomimetics as an integral part of product engineering may remove obstacles for the uptake of biomimetics in general, as it relies on existing well-known design processes.

In biomimetics, communication across disciplines has been discussed several times as a challenge which must be overcome [1,25]. The expert panel needed to find a common language within the committee itself and especially for the description of the biomimetic approaches in the technical standard. Therefore, it was also decided to use and integrate definitions and terms from classical engineering design for the biomimetic process. Specific biomimetic definitions were defined by consensus of the expert panel, while common technical terms were adapted from the existing standards. As such, the biomimetic development process adjusts to the engineering process, and this might further remove inhibitions. In the past, the motivation for describing biomimetics was meant first, to raise awareness so that users are motivated to consider this approach at all, and second, to simplify the various steps which need to be performed to make it easier to understand. This simplification led to a lack of preciseness concerning the usage of specific terms, while in contrast, classical product development reflects a clear terminology and understanding of its procedures. Therefore, using a common language is assumed to make the biomimetic process easier, possibly increase its usage. The definitions of the terms *bionics* and *biomimetics*, which are strictly used in the VDI standards, may be discussed anew, in the future, because these two terms are increasingly being used synonymously.

The work of the VDI 6220-2 committee was an inter- and transdisciplinary endeavor, including academics and practitioners from industry. The aim was to achieve a consensus in understanding the process itself and to describe a process that is oriented towards industrial uptake and application. This occurred by taking the needs and expectations of the industry into consideration, since their representatives were part of the panel.

Having such a large and diverse expert panel may cause problems. Despite careful selection of the interested parties, stakeholders may join the committee later during the working process of developing a standard. In the early phase of the development process, new ideas or even fundamental decisions are introduced and discussed. The introduction of new ideas in the later phases of standard development often impedes the process and should therefore be avoided by involving all interested parties from the beginning, or considering these new ideas again during the objection phase.

For this reason, it often makes sense to start the standardization work towards a technical standard at the national level and to introduce the result into international discussions as a national consensus. The general development process in Germany is highly influenced by the engineering sciences, and the focus of biomimetics is mainly on technical innovation. In other countries, the focus of biomimetics or its usage may differ; for instance, the focus might be primarily towards sustainable development. It would be interesting to see how other countries would set up such a standard and encourage international discussions that might be considered in future revisions of the presented VDI standard.

An advantage of a standard compared to a scientific publication is that its content is based on consensus. However, the development of a new standard requires at least three years of work before publication, and often more. Therefore, even if the state of the art is defined by a standard, it often lags behind the latest developments. Therefore, a standard must be checked regularly to ensure that it is up to date and, if necessary, it requires revision. At the VDI, revision happens every five years to ensure that the standard is adapted to the latest developments [35].

## 5. Conclusions

The main objective of the interdisciplinary expert panel developing the new standard VDI 6220 Part 2 was to provide engineers who are familiar with the classical product development standards with support for the development of biomimetic products. As usual for new standards, there is not yet an experience of how the VDI 6220 Part 2 has been received. The assumption that VDI 6220 Part 2 contributes to improvement must be evaluated in practice. Therefore, subsequent work will need to study how engineers and/or biologists are able to work with this new standard and whether there is an increase in marketable biomimetic products. Research on the acceptance of the new VDI Standard in the classical technical product development process, as well as its application in the development of new biomimetic products, will therefore be carried out in follow-up studies. If an international standard on the methodology of biomimetics is established in the future, VDI 6220 Part 2 will be discussed internationally, and further approaches could be incorporated in an ensuing revised version.

## Figures and Tables

**Figure 1 biomimetics-07-00036-f001:**
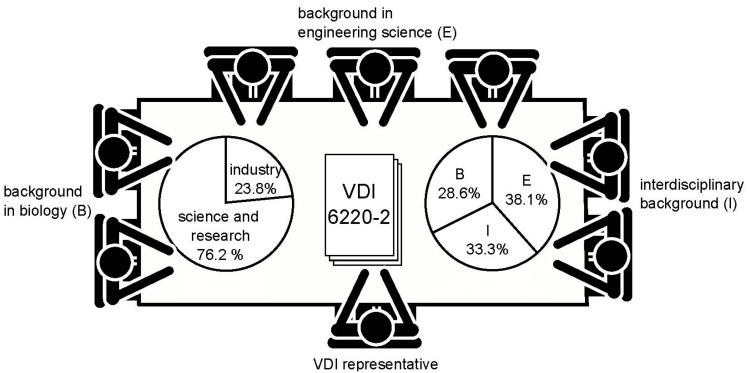
Participants in the expert panel. B: biology; E: engineering; I: interdisciplinary. In total, 21 members, one guest, and one secretary collaborated on developing a process description for biomimetics.

**Figure 2 biomimetics-07-00036-f002:**
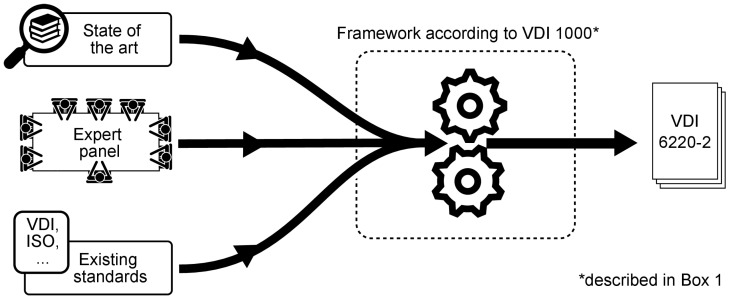
Flow diagram of the working process that led to VDI 6220 Part 2.

**Figure 3 biomimetics-07-00036-f003:**
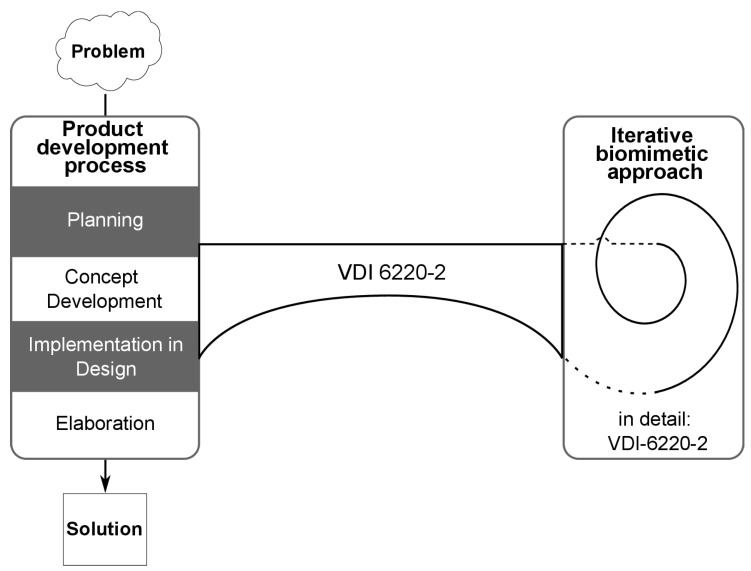
The biomimetic process of VDI 6220 Part 2 linked to the classical product development, for example, as in VDI 2221 Part 1 and Part 2.

## Data Availability

Not applicable.

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
