# Peer review of "Biomimetics Linked to Classical Product Development: An Interdisciplinary Endeavor to Develop a Technical Standard"

_biomimetics, 2022, doi:10.3390/biomimetics7020036_

Round 1
Reviewer 1 Report
It was necessary to have available such a multi-author reference, to intrigue discussions on bio-inspiration, biomimetics or bionics - what ever term we choose. The manuscript nicely describes the background of the standard developed. My doubt is, whether engineers in industry (who are, I suppose, one of the the main groups such contributions aim at - I address the old, complicated relation between theologians, believers and non-believers) due to it's length, multitude of details and orientation towards an underlying "philosophy of design" will find the time and motivation to read and understand it in detail. But for the biomimetics community, it's a valuable base for future discussions.
Reviewer 2 Report
The authors present the paper "Biomimetics linked to classical product development: An interdisciplinary endeavor to develop a technical standard".
The paper is well written but the technical contribution is not clear for a scientific paper.
Moreover I have some comments in order to improve the paper:
Abstract:
It is very strong to affirm that "Biomimetics is a well-known approach for technical innovation. However, most of its outcome remains in the academic field." I disagree, a lot of industrial products can be cited as designing from biomimetics.
All the Keywords used have to appear in the abstract.
Introduction:
"The amount of scientific publications in the field of biomimetics has
strongly increased from less than a hundred in the early 1990s to more than 2500 in 2017 [2, 3]." This number of 2500 in 2017 is not very expressive, what about today? A quick search in Electrical Engineering and we have more of 10 thousand results...
More recent references are requested, also in English
Reviewer 3 Report
Summary:
Overall, the paper is clearly written with clear objectives, methods and conclusions.
Efforts to formalize and advance biomimetic design processes are needed to help translate biomimetic knowledge and insight for the benefit of industry and society. This is a well accepted premise and objective in the field of biomimetics.
Many academic researchers have sought to “crack the code” or advance the success rate of these efforts. Despite the increasing success of biomimetics, BioM or BID outcomes, a more standardized process has yet to be formalized, and I believe this paper is important for simply asking this question and focusing on this goal, by connecting it with existing formal engineering/ product development processes (VDI Standard).
Many of the critical questions or challenges that have prevented Biomimetics from succeeding have been raised as well (Lines 296- 311) namely, problems of scale translation, manufacturability and others.
General Review Comments:
It would be helpful to mention efforts to systematize the search of relevant biological knowledge. These efforts might be cited in the section “The Classical Approach to Product Development….” Starting line 240 to build the author’s argument and widen the breadth of work preceding this study. Goel et al. for example developed the CBID DANE database that allows engineers and designers to search for biological insight according to function. Similar known efforts like the Biomimicry Institute’s “Ask Nature” data base have attempted to do the same. Müller et al. “Bio diversifying Bioinspiration” have suggested how museums of natural history can serve as nimble resources in formal searches of biological insight, but no formal studies have tested or evaluated these potentials. All of these efforts I believe have acknowledged the challenges involved in tagging biological phenomenon so it would come up in relevant searches. I believe acknowledging this topic would help make the process of biomimetics more concrete than simply saying as the authors do on line 270:
“One prerequisite of the biomimetic approaches to start the overall process is the willingness to dive into biology or at least to collaborate. ….. The biomimetic approaches benefit from
interdisciplinarity and teamwork.”
Sometime bioinspiration comes from unexpected domains that cannot be planned- so it is then difficult to know who, and from what area of natural science or biology should be on a team engaged in a problem-based approach.
Round 2
Reviewer 2 Report
After re-reading the cover letter and the revised version carefully, I am still not convinced by the technical-scientific contribution of the paper presented. No new argument was advanced by the authors.
